# Preventive Effects of a Single Bout of Exercise on Memory and Attention following One Night of Sleep Loss in Sports Students: Results of a Randomized Controlled Study

**DOI:** 10.3390/bs12100350

**Published:** 2022-09-21

**Authors:** Johannes Fleckenstein, Sina Gerten, Winfried Banzer

**Affiliations:** 1Department of Sports Medicine, Institute of Sports Sciences, Goethe-University Frankfurt, Ginnheimer Landstr. 39, D-60487 Frankfurt am Main, Germany; 2Neurocognition and Action—Biomechanics Research Group, Faculty of Psychology and Sport Science, Center of Excellence “Cognitive Interaction Technology”, Research Institute for Cognition and Robotics, Bielefeld University, Universitätsstraße 25, D-33615 Bielefeld, Germany

**Keywords:** insomnia, sleep deprivation, exercise therapy, mental performance, shift work

## Abstract

Sleep loss is a severe problem in night-shift workers. It causes fatigue and a decrease in awareness that may be counter-acted by exercise. This randomized controlled study of 22 university students investigated the effects of exercise to prevent loss of cognitive and physical performance following sleep deprivation. We compared a single bout of 20 min circuit training to control in an experimental setting of overnight sleep loss. Outcomes included memory, cognitive tasks, and physical parameters. The occurrence of false memories was considered the main outcome. Exercise did not exert significant effects on false memories (*p* = 0.456). We could detect a trend to significance (*p* < 0.01) assessing cognitive dimensions, i.e., selective and sustained attention, and visual scanning speed. This revealed strong effects of exercise on attention (*p* = 0.091; Cohen’s d = 0.76; ∆14%), cognitive performance, performance speed, and perceived sleepiness (*p* = 0.008; d = 0.60; ∆2.4 cm VAS). This study failed to show the effects of exercise on memory function. Still, the observed effects on attention and consciousness could be considered clinically relevant, as these results encourage further research to determine its practicability and meaningfulness among night-shift workers.

## 1. Introduction

Sleep deprivation and sleep loss negatively impact our quality of life, mood, cognitive function, and health [1]. Insomnia increases the use of healthcare resources and is a risk factor for cardiovascular diseases. Studies show an association between short sleep duration, sleep disturbances, and circadian desynchronization of sleep with adverse metabolic traits, i.e., obesity and type 2 diabetes [2]. Recently an animal study hypothesized that shift-work-induced pathophysiologic changes go beyond the transcriptomic cellular level, regulating the circadian rhythm [3]. As within shift workers [4], bad sleep is common in jobs with high estimated levels of distress, such as truck drivers [5] and railroad employees [6], or jet lag when crossing several time zones [7].

Besides illness, sleep deprivation limits our mental performance. Sleep loss results in the choice of low-effort behavior [8]. In consequence, this reduces the decision-making capability of physicians [9], impairs the cognitive performance of nurses [10], and shooters show degraded marksmanship performance [11]. A study in polysomnography technicians showed one 12 h night shift to be already sufficient to increase perceived sleepiness, reaction time and the number of lapses of attention [12]. Shift work, as investigated in a recent study in nurses, was shown to increase fatigue after day and evening shifts and affect the reaction time after the evening shift [13]. Nurses did not differ in accuracy and maintained a high level of attention among the three working shifts, suggesting a highly developed sense of responsibility in our nurses. A recent study showed that shift work, compared to day work, impairs executive functions, is directly related to shift workers’ poorer sleep and has detrimental effects on critical areas of the brain [14]. Still there was no difference in reaction time. To give a final idea on the dimension of sleepiness, a recent study showed that already five days of sleep restriction induced significant changes on psychomotor vigilance in healthy men [15].

Exercise has been emphasized as a preventive approach to motivate tired populations and to improve the quality of sleep. Among workers with work-related fatigue, 6 weeks of exercise intervention improves employees’ well-being and reduces the degree of emotional exhaustion [16]. This is in line with a trial showing the effects on study-related fatigue among university students [17]. Exercise and physical activity seem suitable to improve the quality of sleep in a variety of specific indications, e.g., pregnancy [18], concerned parents of children with cancer [19], or female university students [20]. There is one study showing 16 weeks of aerobic exercise to improve daytime function and reduce depressive symptoms besides improving sleep in older adults with insomnia [21]. In the elderly, physical activity has been suggested to improve cognition [22] and to protect people from cognitive decline [23]. Finally, the effects of exercise seem dose- and time dependent [24]. Whereas higher intensities mediate less interference and improved reaction time in executive function after 15 min, low- and moderate-intensities have been shown to improve performance after 180 min [24]. In a recent observation from our group, we could show that light physical activity is more likely associated with executive function, whereas maximal workload improves attention in the elderly [25].

In summary, the literature suggests the general benefits of preventive exercise on symptoms following sleep loss [26]. Exercise is easily applicable and would thus be a valuable option for shift workers to be performed after the shift, before being exposed to decision-making and cognitive tasks in their daily life. This exploratory study was designed to determine if a single exercise intervention (in accordance with the literature on higher intensity and time saving) is able to address both functional and mental skills in healthy subjects following sleep loss.

## 2. Participants and Methods

### 2.1. Study Design

A single-center randomized controlled study was conducted, investigating the preventive effects of a 20 min moderate-to-vigorous circuit training on the parameters of cognitive performance when compared to a passive control. The study took place in the summer term 2016 at the Sports Campus of the Goethe-University Frankfurt am Main, Germany. The study was approved by the Ethics Committee FB05 of the Goethe-University of Frankfurt, Germany (reference 2016-23) and is in agreement with the Declaration of Helsinki (Version Fortaleza 2012). The study was registered on 21 June 2016 at the German Clinical Trials Register (DRKS00010655).

Participants checked in at 18:00 the day of the experiment, starting with a collective standardized Mediterranean dinner (grilled chicken, salad, vegetables, and water). Eating was stopped at 20:00 and after two hours of digestion, baseline parameters were assessed at 22:00 (see Figure 1). In the following night, the sleep loss of the participants was assured by the study team. The participants watched two mainstream movies, were allowed to talk, and to play cards or other board games that did not require physical activity or high intellectuality. Slices of vegetables (cucumber, peppers, and tomatoes) were offered till 4:00. At 6:30, participants were randomized into two groups with the “exercise group” leaving to a nearby gym, where they participated in an exercise program, whereas the “control group” remained passively seated in the common area. At 7:30, all outcomes were re-assessed, and the study ended thereafter.

### 2.2. Participants

Participants (>18 <36 years) with good sleep (6 to 8 h/night) and no abuse of substances were assessed for study eligibility using the following exclusion criteria: pregnancy or lactation, intake of neuroenhancers, ability for smoking cessation less than 24 h, suffering from diabetes, severe illness influencing the quality of life (e.g., heart disease, cancer, diabetes, stroke, and arthritis), or severe illness influencing the mental or physical performance (e.g., schizophrenia, and broken leg). All participants signed a written informed consent prior to enrolment.

All participants were recruited through advertising at the campus two weeks prior to the experiment. All participants are students in the field of sports sciences, assuming a physically good condition and daily training load due to the prerequisites of the study program.

Participants were randomly assigned to one of the two study groups using the smartphone-based application, Certified True Randomizers (Integer Generator, Random.org, Dublin, Ireland).

Sample size was estimated using the software G*Power (Version 3.15, University of Düsseldorf, Germany). With α set at 5%, 22 participants were required to have 90% power to detect a decreased number of false memories in the intervention group (estimated 10%) as described for the use of caffeine [28].

### 2.3. Exercise Intervention

The exercise intervention was performed after sleep deprivation at approximately 7:00. Participants exercised together and performed a 20 min lasting circuit training under the guidance of two trainers. First, the warm-up was performed for 5 min and consisted of a series of five movements, each for 60 s, i.e., marching on the spot, side-to-side weight stabilizing, step touches, double step touches, and V-steps. Second, the main exercise was performed for 10 min, including 10 exercises, 30 s each, with a 30 s break—where participants shook their extremities out—in between: (1) air squats, (2) linear lunges, (3) side lunges, (4) push-ups, (5) hip lifts, (6) plank + push-ups, (7) mountain climber, (8) swimming woman, (9) pec fly in push-up position, and (10) bird dog. The warm-up and exercise were accompanied by music at 128 bpm. Third, the cool-down was performed for 5 min including five different moves, 60 s each. Two of the included elements derived from yoga, i.e., sun salutation and the warrior, and the other three were rotation of the vertebra, and stretching of the chest and quadriceps muscles. An internal assessment of this program among Master students of exercise physiology prior to this study revealed the exercise to be of vigorous intensity (60–80% HRR).

### 2.4. Outcome Measures

Demographics included information on smoking, drinking behavior (alcohol, coffee, caffeine, and energy drinks), the extent of physical activity, and the quality of sleep (Pittsburgh Sleep Quality Inventory PSQI, German version) [29]. The PSQI ranges from 0 to 21 with higher scores indicating a decreased quality of sleep, and a PQSI > 5 is a cut-off value for good and bad sleepers.

## 3. Main Outcome Measure

Based on the sample size calculation, the rate of false memories dating from the Desse, Roediger and McDermott (DRM) paradigm [27] was chosen as the primary outcome and implemented as an experimental false memory task by Diekelmann and colleagues [28]. Participants learned 18 DRM lists the evening prior to sleep loss. Each list consisted of 15 semantically associated words (e.g., apple, vegetables, orange, and juice). The list words were presented sequentially by a human female voice with a delay of 10 seconds between lists and 750 ms between words, at the end of the baseline assessments in the evening previous to sleep deprivation. Retrieval took place the next morning in the common area; 108 words were projected in white letters on a black background at the wall. Half of these words originated from the evening before, 36 words were distractors not matching the list topics, and 18 words were theme words that had never been presented before (so-called false memories). Participants gave an old/new judgment for each word (i.e., to indicate whether the word had been presented during learning or not), a confidence rating for their answer on a 4-point scale ranging from 1 (‘‘I had to guess’’) to 4 (‘‘absolutely sure’’), and a remember/know/guess judgment for words judged as “old”. The ratios of false memories, true recognition and false alarms were calculated according to Diekelmann [28].

## 4. Secondary Outcome Measure

Secondary outcomes included psychologic as well as physical parameters. The D2 test of attention was used to assess selective and sustained attention, and visual scanning speed [30]. It consists of a general attention score (total errors per total task, in %), with lower scores indicating higher performance. Subindices include performance speed (total number processed, maximum 299), and concentration performance (speed minus error count), with higher scores indicating improvement.

The severity of sleep deprivation was assed using the Stanford Sleepiness Scale, ranging from 0 to 10 cm (with 0 cm being “fully awake”, and 10 cm “sleeping”) [31].

The German version of the Multidimensional Mood Questionnaire (MDBF; [32]) was used to assess three different dimensions of stress and anxiety, i.e., good–bad mood, awake–tired and calm–nervous, which are supposed to be expressions of mood, mental fatigue, and feelings of restlessness. Higher scores indicate a positive attitude.

Physical parameters included grip strength using a manual dynamometer (JAMAR, Homecraft Ltd., Nottinghamshire, UK), pressure pain threshold (PPT) with a mechanical pressure algometer (pdt, Rome, Italy; range 2–20 kg/cm^2^, diameter 1 cm) upon the belly of the transversal trapezius muscle, as well as the balance error scoring system (BESS; [33]). The BESS is a standardized battery investigating balance in three different stances (double leg, single leg, and tandem stance) on first a firm and second a foam surface (range 0–60 points, with 60 indicating worst balance). 

The sequence of assessments was in the order as described above, with the exception that the D2 test and the test for false memories were performed as the last two assessments.

## 5. Statistical Analysis

Statistical analysis was conducted for comparison of the primary and secondary outcome measures between the two study groups. No evidence was found that the parametric tests used were inappropriate. Baseline characteristics and single time point data were analyzed with unpaired *t*-test (for continuous measures) and chi-square test (for nominal data) to assess for differences among the two study groups. Longitudinal data were analyzed using a time x group repeated measures ANOVA (2 × 2), followed by an unpaired *t*-test of change scores for post-hoc comparisons. Effect sizes according to Cohen were calculated for all relevant outcomes. Significance with α < 10% was considered showing a statistical trend, whereas <5% indicated significant differences. Data are presented as mean ± standard deviation.

Data analysis was performed with the SPSS statistical software system, version 24.0 (SPSS Inc., Chicago, IL, USA).

## 6. Results

Twenty-two participants (8 female and 14 male, age 29.0 ± 3.0 years, weight 71.6 ± 10.7 kg, height 175.6 ± 8.7 cm) were included in the study, and no dropouts occurred during the trial. After randomization, we could observe in each group three left-handed participants, and three participants with a PSQI score > 5 (range 6–11) indicating mild sleeping disorders. There were no differences between groups regarding the intake of alcohol, coffee, caffeine products, energy drinks and cigarettes. Time spent with exercise was similar in both groups. Handedness revealed no group differences. There was no significant difference of any outcome between groups at baseline (Table 1).

We could not detect significant differences between intervention and control regarding the occurrence of the false memory rate (0.80 ± 0.12 vs. 0.76 ±0.16, unpaired *t*-test *p* = 0.456), false alarm rate (0.40 ±0.12 vs. 0.43 ±0.08, *p* = 0.40), or true recognition rate (0.70 ± 0.13 vs. 0.67 ± 0.15, *p* = 0.56, see Table 2). Still, the effect sizes between groups, i.e., false memories (Cohen’s d = 0.32, effect size correlation r = 0.16), false alarms (d = 0.36, r = 0.18), and recognition (d = 0.25, r = 0.12), can be considered small.

Time × group analysis of the D2 test showed a trend (*p* < 0.1) toward improved cognition. Post hoc *t*-tests for inner-subject effects sustained this trend for all D2 items in the exercise group, i.e., attention (*p* < 0.001), as well as concentration performance (*p* < 0.001), and performance speed (*p* < 0.001). The attention score improved within the exercise group by 17.22 ± 15.21% versus 3.53 ± 20.57% in the control group (Table 2, Figure 2). The magnitude indicates a strong effect (d = 0.76, r = 0.35). This effect was consistent for performance (d = 0.78, r = 0.36) and speed (d = 0.71, r = 0.34).

Time × group analysis of sleepiness, the dimensions of mood, balance, grip strength, and pressure pain revealed no significant effects (see Table 2). 

There were inner-subject effects on sleepiness (*p* = 0.019), with an increase of 0.59 ± 4.30 cm VAS in the intervention and 2.97 ± 3.54 cm VAS in the control group, with no significance between groups (post hoc unpaired *t*-test *p* = 0.171). The magnitude corresponds to a strong effect (d = 0.60, r = 0.29).

Subgroup analysis dependent on a PSQI > 5 (cut-off for sleeping disorders) did not alter the observed trends. However, improvements in attention in the exercise group were significantly more pronounced in participants with a PSQI ≥ 5 (∆−32.43 ± 18.29%) versus PSQI < 5 (∆−11.51 ± 32.43%; *p* = 0.033). The PSQI did not affect other outcomes.

Covariate analysis including gender did not reveal differences in the time x group analysis. Female gender led to decreased errors in the balance test (∆−8.25 ± 5.44 vs. ∆+2.71 ± 7.27, *p* = 0.029) in the intervention group. The pressure pain threshold was significantly reduced in female participants in the exercise group (*p* = 0.038), as was the grip strength in both female groups, exercise (*p* = 0.004) and control (*p* = 0.011). Gender had no effects on attention and false memories.

No harm was observed within this study.

## 7. Discussion

Our study indicates the strong preventive effects (Cohen’s d > 0.7) of a single exercise intervention on the D2 cognitive and attention scores as well as sleepiness following one night of sleep deprivation. The power of these assumptions is restricted due to the preliminary character of the study, and as the global statistical test did only indicate trends but not a statistical difference based on an α-error < 0.05. The study was designed in line with the sleep loss experiment described by Diekelmann [28], outcome measures were all established and validated, and we controlled for important influencers such as the incidence of sleeping disorders and the standardization of meals.

### 7.1. Generalizability of Effects

The experimental induction of sleep deprivation was successful, as demonstrated by an increase in the sleepiness scale in the control group by 3.0 points. This in line with other experimental set ups [28,34]. Perceived sleepiness in the exercise intervention group did not relevantly increase (0.59 points). Although this corresponds to a strong effect size (Cohen’s d = 0.6) favoring the exercise group, we could not detect statistical significance. From a clinical perspective, our results are comparable to a study showing exercise to significantly decrease feelings of lethargy, a sensation that might be similar to sleepiness [35].

Exercise showed small effects on memory, but without being significant. It is doubtful that this effect is of clinical importance. Sleep is the best remedy for memory consolidation [36].

In contrast, the effects on cognitive attention in our paradigm indicate strong influence by exercise and are of clinical relevance (mean improvement 15% in attention). One night on sleep deprivation is suggested to rather impact implicit but not explicit learning of serial response time tasks [37]. This may be corroborated by our findings in the D2 test, suggesting a higher performance speed. However, reduced errors as reflected in the D2 global score as well as an increased concentration performance could indicate that acute exercise-induced effects on memory and cognition are not only limited to this improved automatization. This is in accordance with a recent study showing acute exercise to improve executive cognitive functions in young adults [38].

Grip strength was assessed according to current principles, and the baseline force of the dominant hand of women (31.0 ± 4.0 kg) and man (48.2 ± 6.9 kg) was in agreement with German normative reference values [39]. Our data suggest an increase (2 kg) of maximal hand-grip strength. Even though grip strength measures were validated to be robust in regard to test–retest settings, we cannot exclude the effects of habituation. Still, an increase in grip strength could only be observed within the exercise participants; in the control group, grip strength was reduced. To date, no link between sleep loss and grip strength has been reported. Consequently, the effects of exercise on grip strength following sleep loss might be possible.

Sleep deprivation mediates a decrease in pressure pain threshold in both groups, which is an indicator of facilitated pain perception. This is in line with previous findings of sleep disorders and pain sensitivity [40]. Training studies have shown that acute resistance exercise [41] as well as 12 weeks of aerobic exercise [42] are able to increase the pressure pain thresholds in general, effects that we could not confirm following sleep loss. 

We did not detect differences in mood as assessed by the MDBF. Our intervention phase may have been too short to cause significant effects. There is one study reporting a single session of 40 min aerobic exercise (70% VO_2max_) to cause improvements in morning subjective mood states [35]. A study investigating a multimodal 8 month program in caregivers including exercise sessions could observe positive effects on stress and negative affection [43].

### 7.2. Implications of the Exercise Program

The participants in our study performed a single bout of exercise. Studies dealing with the effects of acute exercise on cognition have usually not been performed adding sleep-loss as an additional factor. As described above, Lluch et al. found acute exercise to improve mood and motivation [35]. It has been reported that an acute bout of exercise improves the performance on an executive function task in individuals with schizophrenia [44]. High-intensity resistance exercise is more likely to reduce interference and improve reaction time in the evaluation of executive functions immediately after exercise, whereas low (40%) and moderate (70%) intensity improves performance on plus–minus tasks three hours after exercise [24]. The plus–minus task is a test for switching attention and is slightly comparable to the D2. As we did not assess the 3 h follow-up period, we are not aware if the effects of a single bout of vigorous exercise were even more pronounced at this time. An extended follow-up measurement should therefore be considered for future studies. A meta-analysis suggests acute bouts of exercise to reduce state anxiety [41], an outcome we did not assess in our study.

The mechanism behind the effects of exercise remains unclear. A recent study proposed preliminary data that exercise induces an ‘adrenaline rush,’ this alerting the sympathetic nervous system, triggering a heightened state of alertness [45]. The same authors found a relationship to the cortisol response as well, with greatest effects for high-intensity exercises [46]. The regulation of the vegetative tone by means of regular exercise was also proposed, being a mechanism when dealing with exercise therapy to improve chronic insomnia [47]. In view of the cognitive domain, the results obtained in elderly patients suggest that aerobic exercise elicits neuroprotective metabolic effects in hippocampal structures [23]. Still, these effects are based on continuous exercise protocols.

Finally, the timing of the exercise intervention possibly had an impact on the observed motor skills. It might be possible that exercise elicits greater effects when followed shortly after sleep, and not after sleep deprivation [36].

### 7.3. Limitations

We are aware that the lack of significances between groups limits the generalizability of our findings. Effect sizes are a well-established method to detect meaningful group differences, and the effects found appear quite impressive. In addition, with the small sample size and the character of the study design, one should be aware that findings are only hints and will need confirmation. Furthermore, we investigated the effects in students in physically good condition, a population not directly matched with night shifters.

The DRM paradigm assumes our main outcome to reveal false memories. From a clinical point of view, one could say that it is word recognition to be assessed and, consequently, memory errors. Kathy Pezdek addressed this fact stating that “*The DRM task is a fail-safe semantic priming task that always produces memory intrusion errors. If these memory intrusion errors are equated with false memories, then these findings can be used as evidence for a high prevalence rate for false memories. Unfortunately, however, it has not been demonstrated that the mechanisms that operate to explain the DRM findings apply as well to memory for planting entirely new events in memory*” [48].

As detailed above, this study did not cover the complete range of psychological cognition tests. It was designed to obtain a first insight into a complex understanding. Our study only observed one sleepless night in healthy and physically fit participants, but we did not extend sleep loss to several nights to display more realistic scenarios of daily life.

In addition, we did not control for cardiovascular fitness, as all of our participants were young academics at the sports campus, and their hand grip—correlating with oxygen uptake—was in the expected age-related group. However, fitness may moderate the interaction of acute exercise and cognition [49], and should be assessed on an individual level. Another shortcoming might be the possibility that some of the participants had taken daytime naps, which have been shown to influence declarative memory performance and motivation [50]. Finally, the times of assessment may influence the findings, as cognition and attention underlie circadian rhythms [51].

## 8. Conclusions

Acute circuit training following sleep loss exerts small effects on attention and consciousness that could be clinically relevant when developing preventive health programs in states of sleepiness. These observations are based on a single bout of exercise in an acute overnight stay, and it remains unclear whether they are meaningful for chronic states, e.g., night-shift worker, or not. On the basis of this study, it will be possible to perceive larger and longer-lasting prospective studies to evaluate the preventive health impact of continuous exercise on physical and mental outcomes.

## Figures and Tables

**Figure 1 behavsci-12-00350-f001:**
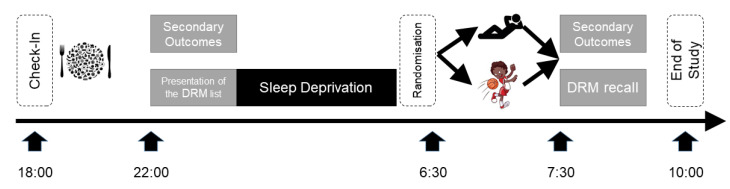
**Study Design.** A single-center randomized controlled study investigating the effects of a 20-min moderate-to-vigorous circuit training (see icons) on parameters of cognitive performance when compared to a passive control. Participants checked in at 18:00 the day of the experiment, starting with a collective standardized Mediterranean dinner. Baseline parameters were assessed at 22:00, and thereafter the list of Desse, Roediger and McDermott (DRM) words was presented [27]. In the following night, sleep loss of the participants was assured by the study team. At 6:30 participants were randomized into two groups with the “exercise group” leaving to a nearby gym where they participated in an exercise program, whereas the “control group” remained passively seated in the common area. At 7:30 all outcomes were re-assessed, the DRM word were recalled, and the study ended thereafter.

**Figure 2 behavsci-12-00350-f002:**
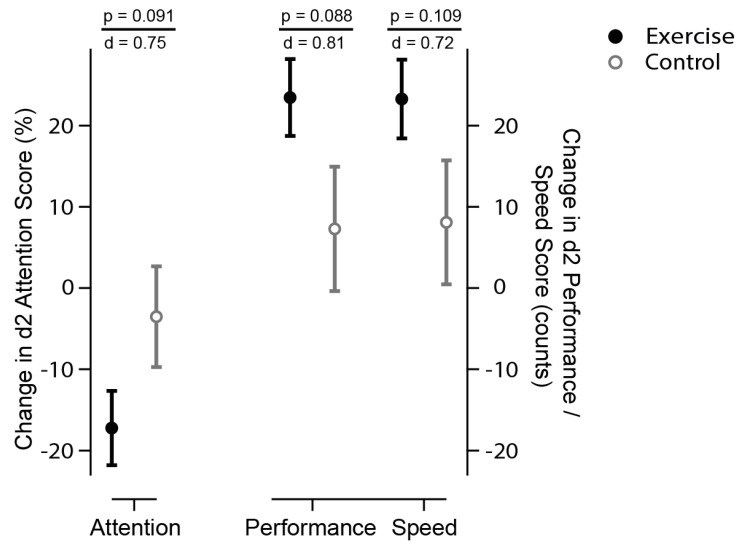
**Selective and sustained attention (D2 Score).** The circles display the changes to baseline for exercise (solid black) and control (open grey circles). The D2 test of attention consists of a general attention score and subindices, including concentration performance and performance speed [30]. It consists of a general attention score (total errors per total task, in %), with lower scores indicating higher performance (left scale). Subindices (right scales) include performance speed (total number processed, maximum 299), and concentration performance (speed minus error count), with higher scores indicating improved performance and speed. There is a statistical trend (time x group analysis, *p* < 0.1) that exercise (solid circles) improved attention (less errors), speed (more numbers processed), and performance (count of processed numbers minus errors) then control (open circles). *p* indicates the level of significance, d represents the effect size according to Cohen, suggesting that a d = 0.2 represents a ‘small’ effect size, d = 0.5 represents a ‘medium’ effect size and d = 0.8 represents a ‘large’ effect size.

**Table 1 behavsci-12-00350-t001:** **Demographics**. Data are reported as mean ± standard deviation, or in counts (n). Statistics between continuous variables was calculated with unpaired *t*-tests (all parametric), and between categoric variables with chi-square tests.

			Exercise Group (n = 11)	Control Group (n = 11)	*p*-Value
*age (years)*			23.64 ± 1.8	24.18 ± 1.89	0.496
*gender (female/male)*			4/7	4/7	0.670
*height (cm)*			175.91 ± 9.51	175.18 ± 8.35	0.702
*weight (kg)*			72.27 ± 9.45	70.91 ± 12.3	0.851
*BMI*			23.37 ± 2.63	22.95 ± 2.35	0.774
*lifestyle factors*					
	alcohol (yes/no)		11/0	10/1	1.0
		irregularly	1	1	0.663
		1 days/week	7	8	
		2 days/week	2	1	
		3 days/week	1	0	
		4 days/week	0	1	
	caffeine products (yes/no)		3/8	3/8	0.682
	coffee	never	7	3	0.554
		<1 cup/day	1	2	
		1 cup/day	1	2	
		2 cups/day	2	2	
		amount (l)	0.28 ± 0.2	0.25 ± 0.05	0.795
	caffeine-containing soft drinks	never	5	7	0.670
		<0.5l/day	6	4	
	energy drinks	never	8	11	0.214
		<1 tin/day	3	0	
	nicotine	none	10	11	1.0
		<5 cigarettes	1	0	
	training activity	hours/week			
		2–3	2	0	0.552
		4–5	3	5	
		6–7	2	3	
		8–9	2	2	
		10–11	0	0	
		>11	2	1	
*Handedness (left/right)*			3/8	3/8	1.0
*PSQI (0–21 points)*			4.45 ± 2.66	4.64 ± 1.63	0.849

**Table 2 behavsci-12-00350-t002:** **Outcome Measures.** The table depicts all outcome measures between the exercise intervention group (EXE) and the control group (CON) at baseline and following one night of sleep loss. The main outcome measure was assessed only once, following sleep loss, applying an unpaired *t*-test between groups. All other statistics was performed as time × group (2 × 2) analysis. Inner-subject effects are displayed calculating paired t test within the groups. VAS, visual analogue scale; MDBF, multidimensional mood questionnaire; BESS, balance error scoring system; D, dominant side; ND, non-dominant side; PPT, pressure pain threshold.

		At Baseline	Following Sleep Loss	Between Subject Effects^#^	Inner-Subject Effects
** *Main outcome* **				**unpaired *t*-test**	
*false memory rate*	EXE (n = 11)		0.8 ± 0.12	0.456	
	CON (n = 11)		0.76 ± 0.16		
*false alarm rate*	EXE (n = 11)		0.4 ± 0.12	0.402	
	CON (n = 11)		0.43 ± 0.08		
*true recognition rate*	EXE (n = 11)		0.7 ± 0.13	0.565	
	CON (n = 11)		0.66 ± 0.15		
** *Secondary outcome* **				**time × group analysis**	
*Sleepiness (cm VAS)*	EXE (n = 11)	*3.94 ± 2.98*	4.53 ± 2.32	0.171	0.357
	CON (n = 11)	*4.64 ± 1.83*	7.61 ± 2.64		0.019
*Attention (%)*	EXE (n = 11)	*44.84 ± 31.4*	27.62 ± 17.53	0.091	**<0.001**
	CON (n = 11)	*42.02 ± 21.52*	38.48 ± 25.4		0.581
*Concentration performance (n)*	EXE (n = 11)	*214.45 ± 43.65*	237.91 ± 31.46	0.088	**<0.001**
	CON (n = 11)	*214.27 ± 30.3*	221.55 ± 38.01		0.365
*Performance speed (n)*	EXE (n = 11)	*215.73 ± 43.87*	239.00 ± 31.07	0.109	**<0.001**
	CON (n = 11)	*215.91 ± 30.98*	224.00 ± 38.44		0.314
*MDBF mood*	EXE (n = 11)	*11.55 ± 0.69*	11.82 ± 1.54	0.673	0.763
	CON (n = 11)	*11.00 ± 1.55*	10.91 ± 1.87		0.898
*MDBF mental fatigue*	EXE (n = 11)	*11.45 ± 1.21*	10.27 ± 0.9	0.335	0.923
	CON (n = 11)	*10.45 ± 1.63*	10.09 ± 2.21		0.629
*MDBF restlessness*	EXE (n = 11)	*12.09 ± 1.38*	10.82 ± 1.47	0.313	0.311
	CON (n = 11)	*11.00 ± 1.55*	11.09 ± 2.51		0.937
*BESS-Score (0–60)*	EXE (n = 11)	*13.82 ± 9.03*	12.55 ± 5.34	0.940	0.219
	CON (n = 11)	*16.91 ± 7.03*	15.36 ± 8.85		0.555
*Grip strength D (kg)*	EXE (n = 11)	*41.27 ± 9.56*	41.55 ± 11.42	0.699	**<0.001**
	CON (n = 11)	*42.64 ± 11.47*	41.09 ± 12.14		0.306
*Grip strength ND (kg)*	EXE (n = 11)	*43.64 ± 10.89*	43.00 ± 11.16	0.054	**<0.001**
	CON (n = 11)	*44.27 ± 11.71*	39.55 ± 12.09		0.098
*PPT D (kg/cm^2^)*	EXE (n = 11)	*9.24 ± 4.31*	9.05 ± 3.78	0.899	**0.028**
	CON (n = 11)	*8.27 ± 3.19*	7.95 ± 2.7		**0.005**
*PPT ND (kg/cm^2^)*	EXE (n = 11)	*7.38 ± 2.45*	6.87 ± 1.81	0.510	**<0.001**
	CON (n = 11)	*6.27 ± 1.91*	6.33 ± 1.96		**0.005**

## Data Availability

All data can ben obtained upon request from the corresponding author.

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
