# Peer review of "Preventive Effects of a Single Bout of Exercise on Memory and Attention following One Night of Sleep Loss in Sports Students: Results of a Randomized Controlled Study"

_behavsci, 2022, doi:10.3390/bs12100350_

Round 1

Reviewer 1 Report

please see attached document. 

Author Response

We thank the reviewer for his appreciation and agree with the major concern, that observations could be overestimated. We thus carefully revised our manuscript.

As all of our participants were students in sports sciences, physical condition is mandatory, as they have to pass a physical fitness test prior to their inscription, and have to pass several exercise exams every term. Consequently their daily training load is increased. We amended the manuscript accordingly (p.3). The title has been altered accordingly: “Preventive effects of a single bout of exercise on memory and attention following one night of sleep loss in sports students: results of a randomised controlled study”

Ad 1.) We agree that this term is strong, and had the same opinion previously. Still, the false memory task relies on the DRM paradigm. To supplement the idea of an associative model, Roediger and McDermott suggested that retrieval processes might play an important role in explaining the DRM phenomenon. They suggested that, by simply recalling the actual list words, the availability of the lure increases to a point where it is mistaken for a presented word. Critics, however, have argued that the DRM paradigm does not reflect real life events because of the nature of the stimuli and the setting in which the study is conducted. It has been suggested that it is inappropriate to compare the recognition of a word with the implantation of a memory for an entire childhood event (https://doi.org/10.1016%2Fj.concog.2005.06.006). Roediger and McDermott maintain that their use of college students in a laboratory setting with mundane stimuli only strengthens their point, because these conditions should promote the most accurate remembering, and yet false memories are still formed. Still, to address this point we amended the discussion section

Ad 2.) In agreement with the other reviewers we amended the introduction section (p.2) accordingly giving more examples.

Ad 3.) This is an interesting point. As we opted to provide a short-lasting intervention we decided for a higher intensive exercise protocol in agreement with current training principles. This is in line with findings by Brush et al, and we amended the introduction as suggested.

Minor revisions:

Ad 4.) The sentence was a subheading and has been omitted

Ad. 5) changed

Ad 6.) changed

Ad 7.) typo resolved

Ad 8.) resolved

Ad 9.) revised

Ad 10.) agree and comma deleted.

Ad 11.) revised

Reviewer 2 Report

The authors examined the effect of a single bout of a 20-minute exercise on preventing loss of cognitive and physical performance following a full night of sleep loss. While there are studies testing regular exercise training on cognitive performance after sleep deprivation, the authors used a single-section of exercise protocol after the sleep deprivation. The major concerns are the scientific premises of exercise training immediately after a night of sleep loss as well as its clinical implications, particularly for those who work night shifts. Below are a few things the authors may wish to consider, which, in my view, would strengthen the manuscript.

Introduction:

The authors may expand on the literature review on exercise and cognitive function, for example, evidence-based intensity, duration and timing of exercise, the effect of exercise on specific cognitive domains, and more importantly, the effect of exercise on preventing cognitive decline and promoting functional recovery after sleep deprivation. This information is important to justify the research hypotheses, such as the exercise protocol (single section vs. regular exercise, before vs. after sleep loss, et al.) and selection of cognitive domains and timing of cognitive assessment. 

Methods: 

Samples: Participants with good sleep were recruited. However, those with a PSQI>5 (n=3) were not excluded from this study. Could the authors provide clarification on inclusion criteria? Also, could the authors provide examples of severe illness influencing the quality of life/performance (see exclusion criteria)? Were those with cardiovascular diseases excluded?

Protocol: were naps before the experimental study allowed? Daytime naps may compound cognitive changes after total sleep deprivation.

Outcome measures: memory is the primary outcome. Did the authors consider a comparison of memory retention before sleep deprivation? It is unclear whether memory retention between groups is the same at baseline.

Results

Please clarify “Three participants with a PSQI score > 5 (range 6-11) indicating mild sleeping disorders”. A PSQI score of 11 seems high.

Discussion

Could the authors comment on the health impact of exercise immediately after total sleep loss, especially in comparison to the effect of exercise after catch-up sleep on function recovery? What is the clinical implication of timing, intensity, and duration of exercise on preventing cognitive decline following sleep deprivation (i.e., for shift workers)? The authors briefly mentioned some of the information in the conclusion. Could the authors expand on these points in the discussion section?

While the baseline cognition was assessed in the evening, follow-up measurements took place in the morning. Could the authors discuss the time of day effect on cognitive performance, as well as the impact of possible daytime naps before the intervention?

Conclusion

The authors may temper the statements about the effect since the results were not statistically significant.

Author Response

We thank the reviewer for her/his appraisal.

Introduction: In agreement with you and the other reviewers we expanded some of the points, especially the impact of sleepiness, as well as the rationale of exercise interventions

Methods: We agree that this might be confusing. We first decided to avoid all type of sleep disorders, with PSQI > 5 defined as the cut-off. However, as described in a sub-group analyses (p.8) this fact did not change our observations, we decided to analyse all of our data.

The inclusion criteria section has been revised, CVDs and all other diseases requiring constant medical care have not been included

Protocol: We cannot fully exclude whether participants have taken a nap during the day. It was a regular working day, and all participants have been on the campus and involved in their studies during day time. Still, given the possibility, we amended the limitations part of the manuscript addressing this important fact

Outcome measure: We agree this to be an interesting detail. However the DRM paradigm is not set up for repeated measures. We adhered to the study design as proposed by Diekelmann et al.

Results: see above. We analysed data including and excluding those 3 participants. As results remained unchanged we decided to include the 3 subjects to strengthen the power of findings and to avoid a selection bias.

Discussion: We amended the discussion and introduction accordingly (p.10 and 2). For the sake of brevity we gave some ideas, but decided not to end up in a review paper.

Effects of the time of assessment cannot be fully excluded –a fact we amended in the limitations section as well-. Still, as we controlled this study with a no treatment group and did not find a relevant change in neither selective nor sustained attention, we belief this to be marginal.

Conclusions have been revised accordingly

Reviewer 3 Report

Dear Authors, 

the article is well structured and complete in every part. 

The theme is original and clear in its exposition. 

I suggest that you expand the introduction further with more references to literature (if any) in order to better support your results in the discussion section.

I hope that these suggestions can enrich the quality of your article.

Author Response

IWe thank the reviewer for his appreciation and carefully revised both introduction and discussion adding further references supporting the body of evidence.

Round 2

Reviewer 1 Report

The authors have made appropriate adjustments to their manuscript.  One minor comment is that in Table 1  "coffein-containing softdrinks"- caffeine is still misspelled here. 

Reviewer 2 Report

Thank you for the opportunity to review the revised manuscriptThe authors have  addressed my major comments, especially the research evidence supporting their intervention approach.